# Descending Necrotizing Mediastinitis Caused by *Streptococcus pyogenes* in a Child with Primary Epstein–Barr Virus Infection

**Miki Yoshimura** [1], **Tomoo Daifu** [1,*], **Minoru Suehiro** [1], **Tsuyoshi Shoji** [2] **and Yoshihisa Higuchi** [1]

1   Department of Pediatrics, Otsu Red Cross Hospital, Otsu 520-8511, Japan
2   Department of Thoracic Surgery, Otsu Red Cross Hospital, Otsu 520-8511, Japan
*   Correspondence: daifu-kyt@umin.ac.jp; Tel.: +81-77-522-4131

**Abstract:** Descending necrotizing mediastinitis (DNM) is a severe, life-threatening disease with a high mortality rate resulting from sepsis or other complications. DNM can also be a rare and severe complication of Epstein–Barr virus (EBV) infection in adolescents and young adults but has never been reported in a pre-school child. A 4-year-old girl was admitted to our hospital with a 2-day history of fever and chest pain. Computed tomography (CT) revealed a right sided pleural effusion, fluid collection in the retropharyngeal and mediastinal areas, cervical lymphadenopathy, and marked hepatosplenomegaly. She was diagnosed with empyema, retropharyngeal abscess, and mediastinitis. To improve her dyspnea, a chest tube was inserted, and antibiotic treatment was initiated. Her condition improved temporarily, but on day 5 in our hospital, she developed a fever again. A repeat CT scan showed exacerbation of fluid retention in the retropharyngeal area and the mediastinum, for which she underwent drainage and debridement of necrotic tissue in the retropharynx and mediastinum. The presence of cervical lymphadenopathy and marked hepatosplenomegaly suggested the involvement of EBV. Serological tests for EBV revealed primary EBV infection at the time of the DNM onset. Finally, she was diagnosed with DNM following primary EBV infection. At follow-up 1 year later, she was doing well. The risk of DNM should be recognized in patients, even pre-school aged children, with primary EBV infection.

**Keywords:** descending necrotizing mediastinitis; *Streptococcus pyogenes*; primary EBV infection; pediatrics

## 1. Introduction

Descending necrotizing mediastinitis (DNM) is an oropharyngeal infection that spreads to the mediastinum, most frequently following odontogenic infections, peritonsillar or retropharyngeal abscesses, cervical lymphadenitis, trauma, or endotracheal intubation [1]. DNM is a severe, life-threatening disease with a high mortality rate resulting from sepsis or other complications [2], and *Streptococcus pyogenes* (*S. pyogenes*) is one of the causative pathogens of retropharyngeal abscesses and the subsequent rapidly progressive DNM [1,2]. This disease is rarely seen in the pediatric population and its occurrence has been only occasionally reported [3,4], except for DNM caused by methicillin-resistant Staphylococcus aureus in infants and young toddlers [5]. DNM can also be an extremely rare and severe complication of Epstein–Barr virus (EBV) infection in adolescents and young adults [6]. To the best of our knowledge, however, DNM associated with EBV has never been reported in a pre-school child. This report describes a pre-school patient with EBV-associated DNM.

## 2. Case Description

A 4-year-old girl was admitted to our hospital with a 2-day history of fever, chest pain, and agitation. Her past medical history was unremarkable, and she had a negative family history of primary immunodeficiency disease. She had a temperature of 37.9 °C, a heart rate of 190 beats per minute, and an oxygen saturation of 97% in room air. Physical examination revealed clouding of consciousness and lip swelling. Laboratory tests showed a white blood

cell count of $49 \times 10^3$/L (neutrophils, 90.5%, and atypical lymphocyte, 1.0%), a C-reactive protein concentration of 32.6 (normal range: 0–0.5) mg/dL, a lactate dehydrogenase concentration of 434 (normal range: 80–220) U/L, an aspartate aminotransferase concentration of 31 (normal range: 8–35) U/L, an alanine aminotransferase concentration of 20 (normal range: 5–40) U/L, and an anti-streptolysin O concentration of 1908 (normal range: 0–200) IU/mL. Chest X-rays revealed decreased permeability throughout the right lung field, and computed tomography (CT) revealed a right sided pleural effusion (Figure 1a), fluid collection in the retropharyngeal (Figure 1b) and mediastinal (Figure 1c) areas, cervical lymphadenopathy, and marked hepatosplenomegaly (Figure 1d). The patient was diagnosed with empyema, retropharyngeal abscess, and mediastinitis. She required admission to the intensive care unit due to respiratory failure. To improve her dyspnea, a chest tube was inserted, and meropenem treatment was initiated. The pleural fluid was turbid and exudative, with Gram-positive cocci. The culture of the fluid was positive for *S. pyogenes*. The patient's condition improved temporarily, but on day 5 in our hospital, she developed a fever again. A repeat CT scan showed exacerbation of fluid retention in the retropharyngeal area and the mediastinum, for which the patient underwent drainage and debridement of necrotic tissue in the retropharynx and mediastinum. In the operation, the otolaryngology–head and neck surgery team performed retropharyngeal drainage and debridement with a left neck incision and two drainage tubes inserted, followed by the drainage and debridement of the mediastinal abscess by thoracic surgery team, using a bilateral video-assisted thoracic surgery (VATS) approach, with the two chest drain tubes (24 Fr) inserted on each side. The bilateral VATS procedure was sufficient for the drainage of the mediastinal abscess, including the retrosternal abscess, without using median sternotomy. The drainage tubes in the retropharyngeal space and the bilateral chest drain tubes were extubated on postoperative day 6. After detection of *S. pyogenes*, the patient was started on an intravenous drip of penicillin and clindamycin, after which she continued to recover steadily. The presence of cervical lymphadenopathy and marked hepatosplenomegaly at the time of diagnosis suggested the involvement of EBV. Serological tests for EBV revealed IgM and IgG antibodies specific to the EBV viral capsid antigen, but the patient was negative for IgG antibodies reactive with the EBV nuclear antigen, findings that are indicative of primary EBV infection at the time of the DNM onset. Finally, she was diagnosed with DNM and empyema following primary EBV infection. She had postoperative complications, including a catheter-related bloodstream infection and drug fever, but was discharged after 85 days in our hospital. She underwent immune function testing (immunoglobulins and complements), which revealed no deficits. At follow-up 1 year later, she was doing well.

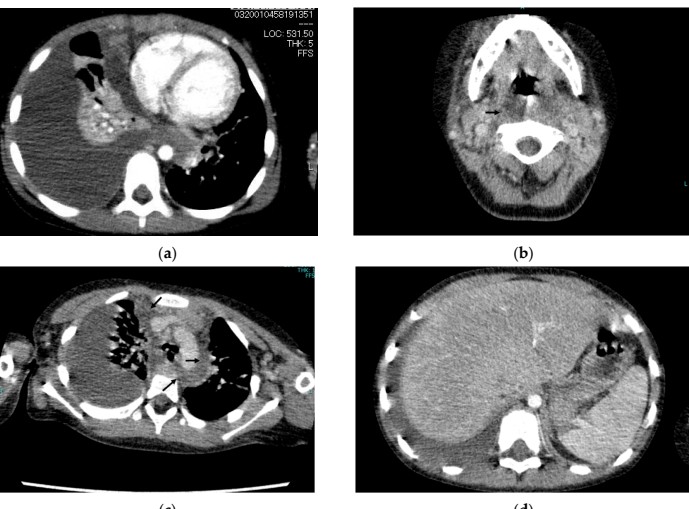

(a)     (b)     (c)     (d)

**Figure 1.** Cervical, chest, and abdominal computed tomography (CT) scans of the patient show a right sided pleural effusion (**a**), fluid collection in the retropharyngeal area ((**b**), arrow), fluid collection in the mediastinal area ((**c**), arrow), and marked hepatosplenomegaly (**d**).

## 3. Discussion

Although DNM is a very rare disease, it is clinically important because of its high mortality rate. Early diagnosis is very important, and aggressive surgical drainage of the mediastinal abscess is recommended for successful treatment [1]. The patient in this case was 4-year-old girl, and selective, one-lung, mechanical ventilation using a double-lumen endotracheal tube during surgery under general anesthesia was not a viable option because of her small bronchial diameter. However, the drainage and debridement of the mediastinal abscess using a bilateral VATS approach was sufficiently possible and effective as a less surgically invasive treatment, even under bilateral mechanical ventilation.

Two children with DNM caused by *S. pyogenes* have been described to date [3,4], and the clinical course of DNM caused by *S. pyogenes* is characterized by severe systemic toxicity and alarmingly rapid progress, indicating the need to start appropriate treatment as soon as possible. Interestingly, one of these patients had chickenpox as a comorbidity [4]; similarly, the patient described in this study was co-infected with EBV. *S. pyogenes* has been identified as a bacterial pathogen in patients co-infected with viruses, including varicella virus [4], influenza A virus [7], and EBV [8]. Thus, immunosuppression following viral infection may predispose patients to DNM caused by *S. pyogenes*.

To date, several adolescents, and young adults aged 17–39 years, have been reported with having DNM related to EBV infection [6]. EBV is thought to induce immunosuppression, with a transient decrease in T cell-mediated immunity, which may predispose patients to bacterial super-infection [6]. In our case, we were not able to evaluate the immune status of the patient at the time of the onset of the disease, and further investigation is needed to determine the mechanism of pathogenesis. Generally, the incidence of serious complications in patients with primary EBV infection increases with age and is rare in young children. However, the findings in the present patient indicate that severe bacterial infections can occur in children, even those of pre-school age.

In summary, to the best of our knowledge, this patient was the first pre-school child to be diagnosed with DNM and primary EBV infection. The risk of severe bacterial infection, including DNM caused by *S. pyogenes*, should be recognized in patients, even pre-school aged children, with primary EBV infection.

**Author Contributions:** M.Y. and T.D. managed the patient and wrote the manuscript. M.S., T.S., and Y.H. managed the patient and revised the manuscript. All authors have read and agreed to the published version of the manuscript.

**Funding:** This research received no external funding.

**Institutional Review Board Statement:** Not applicable.

**Informed Consent Statement:** Informed consent was obtained from the patient's family for the publication.

**Data Availability Statement:** Not applicable.

**Conflicts of Interest:** The authors declare no conflict of interest.

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
