# Peer review of "Descending Necrotizing Mediastinitis Caused by Streptococcus pyogenes in a Child with Primary Epstein–Barr Virus Infection"

_pediatrrep, doi:10.3390/pediatric15010003_

Round 1

Reviewer 1 Report

Dear Editor and Authors,

Thank you for asking me to review this very interesting case report by Dr. Yoshimura and colleagues from the Otsu Red Cross Hospital in Otsu, Japan titled “Descending necrotizing mediastinitis in a child with primary Epstein-Barr virus infection” in which they describe the case of a 4 year old pre-school child which developed a serious descending necrotizing mediastinitis (DNM) post Epstein-Barr virus (EBV) infection and which was managed by them quite successfully I may add.

As a practicing thoracic surgeon the earliest similar case I have encountered was a 15 year old boy and this was caused not by EBV but by a descending staph. infection from a tooth abscess. Therefore, this case has for sure rarity and interest. The manuscript is well written in clear and understandable language with only minor English language corrections needed. It is also well illustrated with radiological images although some surgical pictures if any are available would enhance it more.

I have some minor comments to make:

1. I would change line 48 and instead of saying “fluid retention” I would suggest “revealed a right sided pleural effusion (fig. 1a) and a fluid collection in the retropharyngeal (fig. 1b) and mediastinum (fig. 1c) areas”

2. The authors should mention if the pleural fluid drained following drain insertion was clear or turbid/pussy! They should also mention it was an exudate and that bacteria were seen (or not) in microscopy.

3. The thoracic surgeon on the team of authors (Dr. Shoji) should describe in more detail the surgical approach and findings of the surgery. Was a median sternotomy was performed to drain the retrosternal abscess? How was the retropharyngeal collection drained? How many drains (if any) were inserted and when were they removed?    

4. Did the patient require PICU hospitalization?

5. Why was the length of stay 85 days!!! Were there any post-operative complications?

The discussion is good although as a surgeon I would like to know a bit more about the options of surgical management/drainage/procedures one can use.

In conclusion, I am positively predisposed towards this case report as it does have merit for publication following the authors addressing some minor comments. Thank you again for asking me to review this case report. I wish all well and good job to the authors.

Kind regards,

Reviewer 2 Report

The paper is clearly written and reads easily. I have however some comments to make:

- The title is misleading as if DNM is caused by EBV whereas the causative pathogen was a S.pyogenes. This needs to be changed.

- In the introduction, there is no mention of DNM and S.pyogenes.

- Case description: were there any EBV-PCR samples taken to confirm active infection? did the authors perform an immune screen in the patient described (T-cell mediated immunity)?

- Discussion: there is no additional evidence in the case report to suggest what the authors discuss, namely that the EBV infection causes immunosuppression leading to the DNM by S.pyogenes.

minor:

- arrows on CT-images to highlight was needs to be seen

- what does the author mean by 'severe' opacities (line 47); I would rather describe the extent

Reviewer 3 Report

The authors present an interesting case of a patient infected with EBV who developed descending necrotizing mediastinitis. The study is interesting, but it contains several errors. The case description is quite superficial.

The work requires some adjustments. In places imprecisely made, e.g.: no standard for CRP, LDH written with a capital letter for an unclear reason, Some pathogens are written in italics and some in normal font.

"Generally, the incidence of serious complications in patients with primary EBV infection increases with age, with severe complications being rare in young children."  - the sentence needs improvment. The repetition "complications' needs correction. 

Round 2

Reviewer 2 Report

Changes were made appropriately.

Reviewer 3 Report

Thank you. Reviewer's suggestions included